# StarStream: Live Video Analytics over Space Networking

## ABSTRACT

Streaming videos from resource-constrained front-end devices over networks to resource-rich cloud servers has long been a common practice for surveillance and analytics. Most existing *live video analytics* (LVA) systems, however, are built over terrestrial networks, limiting their applications during natural disasters and in remote areas that desperately call for real-time visual data delivery and scene analysis. With the recent advent of space networking, in particular, Low Earth Orbit (LEO) satellite constellations such as Starlink, high-speed truly global Internet access is becoming available and affordable. This paper examines the challenges and potentials of LVA over modern LEO satellite networking (LSN). Using Starlink as the testbed, we have carried out extensive in-the-wild measurements to gain insights into its achievable performance for LVA. The results reveal that, the uplink bottleneck in today's LSN, together with the volatile network conditions, can significantly affect the service quality of LVA and necessitate prompt adaptation. We accordingly develop `StarStream`, a novel LSN-adaptive streaming framework for LVA. At its core, `StarStream` is empowered by a transformer-based network performance predictor tailored for LSN and a content-aware configuration optimizer. We discuss a series of key design and implementation issues of `StarStream` and demonstrate its effectiveness and superiority through trace-driven experiments with real-world network and video processing data.

## CCS CONCEPTS

• **Networks** → **Network measurement**; • **Information systems** → **Multimedia streaming**.

## KEYWORDS

live video analytics, LEO satellite, network measurement

## 1 INTRODUCTION

Live video analytics (LVA) [13–15, 20, 21] analyzes online video data from networked cameras for automated knowledge extraction. Given the limited resources in front-end cameras [14], the video data are typically streamed to resource-abundant edge or cloud servers for real-time analytics [4, 14, 15, 40]. To date, most data are carried over terrestrial networks, which have covered much of the human-inhabited lands and major roads in developed countries. Yet, despite global efforts to connect the uncovered, unserved, and underserved populations to the Internet, one-third of the Earth's population (around 2.9 billion people) remains disconnected [25].

*ACM MM, 2024, Melbourne, Australia*

© 2024 Copyright held by the owner/author(s). Publication rights licensed to ACM.
ACM ISBN 978-x-xxxx-xxxx-x/YY/MM
https://doi.org/10.1145/nnnnnnn.nnnnnnn

Many rural and remote areas, crucial for industries such as mining, fishing, or forestry, have not been covered, or may never be, including places like Alaska in the USA, northern Canada, and central Australia. Additionally, their terrestrial network services can be vulnerable to disruptions from extreme weather, natural disasters, or other emergencies.

To bridge this digital gap, recent years have seen the rapid development and deployment of space networking based on Low Earth Orbit (LEO) satellites. Compared to geostationary (GEO) satellites at the 35, 786 km orbit, LEO satellites are much closer to Earth (below 2, 000 km). This proximity significantly reduces the launching cost as well as the signal travel distance, resulting in substantially lower network latency (600+ vs. 25 ms [23]). With a massive amount of smaller satellites operating in higher frequency bands, an LEO satellite constellation provides truly global coverage with much higher bandwidth than GEO (e.g., 178 vs. 82 Mbps median download throughput [18]). As a key player in this field, Starlink has approximately 5, 788 LEO satellites serving at altitudes about 550 km up to date, with plans to expand to 42, 000 over the long haul [31]. They together offer high-speed and affordable Internet access to 2.6 M subscribers, many in remote areas with no terrestrial accesses [11, 17, 18], making latency- and bandwidth-critical LVA anywhere on the Earth possible.[1]

As the *LEO satellite networking* (LSN) evolves into a seamless global coverage, LVA applications built upon it will accordingly expand to currently underserved areas. With LSN, LVA applications that are difficult to develop with terrestrial networks, such as disaster response and relief, wildlife monitoring, and maritime surveillance, can be easily implemented. Even in urban areas with well-established terrestrial infrastructures, LSN can serve as a resilient alternative or even the preferred communication method for latency and cost reasons. Starlink [22] has recently partnered with cloud providers to install ground stations (GSes) within or near cloud data centers to process user data directly from space at the edge of the LSN [6, 11]. The enhanced integration of communications and computing infrastructures creates new possibilities for realizing resource-efficient LSN-enabled LVA.

Recent measurement studies [11, 18, 43] have confirmed the advantages of LSN over its GEO counterparts and identified its potential in supporting various network-intensive applications, such as video streaming [43] and cloud gaming [18]. However, LSN shows highly asymmetric uplink and downlink performance[2], with the mean download throughput being more than 10× higher than the upload throughput according to our measurements. The scarcity of uplink resources poses non-trivial challenges for LVA, which primarily consumes the upload bandwidth to ship videos from cameras to remote servers. Furthermore, being highly susceptible to environmental factors (e.g., precipitation, cloud cover, and temperature [11, 17]) and relative satellite motion, LSN exhibits wildly

---

[1] Starlink's coverage has not yet reached polar areas with its current orbit inclination of about 53°, but it is expected to cover these regions in the coming years.

[2] In this paper, *uplink* is the transmission path of sending data from a piece of user equipment to a server while *downlink* is the opposite transmission path.

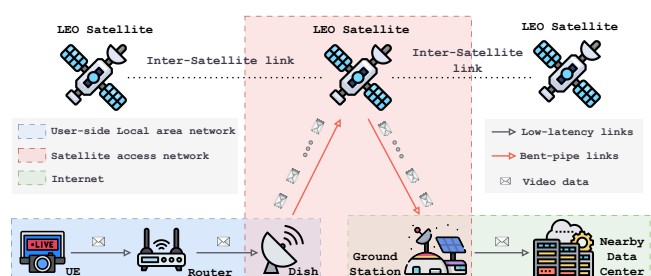

**Figure 1: An overview of LSN-enabled LVA.**

fluctuated performance over time. This prevents upper-layer LVA applications from delivering consistent quality of experience (QoE).

This paper closely examines LSN performance from an end-user perspective and sheds light on building LVA services over the space. We take Starlink as the testbed and discuss the key issues towards designing `StarStream`, a high-performance and analytics-oriented live video streaming framework over LSN. Our contributions can be summarized as follows.

⋄ Based on commercial LSN and cloud computing offerings, we conduct a large-scale measurement study to understand the **access** network performance of LSN. In particular, we identify the scarcity and dynamics of the available uplink resources.

⋄ We further carry out extensive **in-the-wild** measurements to examine the achievable LVA performance of the status quo streaming method over today's LSN. Our findings indicate that achieving high-performance LVA streaming remains a challenge.

⋄ We propose `StarStream`, a novel LSN-adaptive LVA streaming framework, to overcome the unique challenges brought by LSN. Specifically, a transformed-based network performance predictor is designed to adapt the encoding group of pictures (GOP) length to throughput variations while offering throughput predictions for GOP configuration selection. With the predictions, a content-aware optimizer is further developed to strike a good balance between accuracy and latency via configuration optimization.

⋄ Using fine-grained LSN uplink network traces from the real world, we evaluate the performance of the predictor, verifying its advantages over competitors. By integrating real-world collected video streaming traces, we further examine `StarStream`'s performance, and extensive experiments have validated the framework's effectiveness and superiority.

## 2 STARLINK NETWORK ACCESS: PRELIMINARIES AND MEASUREMENT

Figure 1 shows a typical Starlink setup for end users, where a router connects the user equipment (UE) to the LSN, and a dish (antenna) is responsible for communicating with the satellites. Although Starlink has embarked on the deployment of inter-satellite links [29], single *bent-pipe* communication [11, 17] is still the dominant case. For example, according to our thousands of `traceroute` records, there is only one *bent-pipe* communication (one ground-space hop and one space-ground hop) along the path to the destinations, and Starlink tends to serve as an access network to the Internet.

**Table 1: Starlink access network performance (mean ± standard deviation) between the UE and analytics servers.**

| Network performance metrics | gc-server | aws-server |
|---|---|---|
| Download throughput (Mbps) | 83.4 ± 60.5 | 110.1 ± 57.5 |
| Upload throughput (Mbps) | 8.1 ± 3.3 | 8.3 ± 3.5 |
| RTT (ms) | 46.9 ± 14.4 | 40.5 ± 16.4 |

We emphasize that the key differences here, as compared to terrestrial network access, are that no terrestrial cabling or towers are required, and satellites take on the role of relaying data between user dishes and GSes, both of which act as interfaces between terrestrial and space networking. Given the stringent latency requirements of LVA, we are particularly interested in Starlink's access network performance, i.e., the performance of directly consuming data from space in the proximity of GSes. This is in contrast to existing measurements on Starlink Internet access from different global vantage points [11] or from a single vantage point to access globally distributed servers [17, 18]. As such, we set up two servers near the Starlink GS, one from Amazon Web Services, named `aws-server`, and the other from Google Cloud Platform, named `gc-server`. We tried to make the servers as close to the LSN (i.e., the Starlink GS) as possible. Specifically, based on the network latency, servers are rented from the cloud regions with the minimum mean round-trip time (RTT) to the UE. Both servers have Gbps bandwidth for inbound and outbound traffic and will not be the bottleneck of the network performance test. We use `iPerf3` [10] utility to measure the TCP throughput and `Ping` utility to measure the RTT, and the test is executed every 30 minutes.

Table 1 shows the statistical results of **1,056 tests** over **22 days**. As shown, the mean network latencies between the UE and the servers are decent and comparable to recently reported LTE network results (i.e., 47.6 ± 8.4 ms [9]) but with higher fluctuations. Moreover, Starlink follows a *download-centric* design, where the mean download throughput can be more than 10× higher than the corresponding upload throughput for both servers. Unfortunately, LVA differs from traditional human-oriented video streaming applications in that it uses the uplink rather than the downlink to transmit heavy and bursty video data. Consequently, *accommodating LVA can significantly challenge Starlink's asymmetric network link design*. In addition, Starlink's mean upload throughput (≤ 8.3 Mbps) is dramatically lower than that of LTE (53.4 Mbps), let alone 5G mmWave (52.8-131.8 Mbps) [9]. With the upload throughput, it can be challenging for Starlink to live stream Ultra High-definition (UHD, 4K or 8K) videos or multiple Full HD (FHD, 1080p) videos simultaneously. For example, YouTube's recommended live streaming bitrate for a 1080p (4K) video is 3-6 Mbps (13-34 Mbps) [36].

We analyzed the variations in the upload throughput across different days and observed no significant daily patterns. We further divided the measurement results into peak hours (7AM-11PM) and off-peak hours (11PM-7AM) usage and found that, statistically, Starlink provides better network service during off-peak hours than during peak hours. For instance, the mean off-peak upload throughput from the UE to the `aws-server` is 9.2 Mbps, considerably higher than that of peak hours (8.1 Mbps). When we took a closer look

**Table 2: The video dataset used in this paper**

| Name | Source [Link] | Description |
|------|---------------|-------------|
| hw1 | YouTube [35] | Highway traffic camera |
| hw2 | YouTube [37] | Highway traffic camera |
| street | YouTube [38] | Live street webcam in Tailand |
| beach | YouTube [39] | Ocean view from a beach camera |

at the variations over the course of a day, we found that the upload throughput fluctuates wildly. For example, it can be as high as 16.5 Mbps (15.6 Mbps) and as low as 2.2 Mbps (1.9 Mbps) for the `gc-server` (`aws-server`) within the same day. A finer examination revealed that the upload throughput is highly volatile from second to second. For example, the upload throughput for both servers can vary from 0 to 18+ Mbps within a minute. LEO satellites move faster relative to Earth due to their low orbit altitudes, which means that the communication distance and communication link quality between the UE (even a stationary one) and the serving satellite are constantly changing, and frequent handovers from one satellite to another are unavoidable [11, 17]. Hence, we hypothesize that the observed wild performance fluctuations are inherent to the LSN.

## 3 WHEN LVA MEETS LSN: A REALITY CHECK

LVA is recognized as a key technical enabler for a wide range of modern applications, from security surveillance and traffic control to self-driving cars and augmented reality [2]. It relies on advanced learning and computer vision algorithms, such as object detection, semantic segmentation, and human key point detection, to analyze live camera videos, enabling machines to locate, track, classify, and segment video content of interest. The accuracy of LVA is typically preserved by deep neural network (DNN)-based vision models, which are known to be resource-intensive. This popularizes LVA streaming, which involves transmitting live video content from the capturing camera over networks to an analytics server. This is akin to the first-mile ingestion phase of the canonical live video streaming setup where the live content is transmitted from the capturing device to a streaming server [45]. Real-time messaging protocol (RTMP) is the state-of-the-practice ingestion protocol utilized by mainstream commercial live streaming platforms such as Twitch [26] and YouTube Live [36], as well as IP cameras [33] for real-time video upload. Therefore, we use RTMP as the streaming protocol to explore in-the-wild LVA performance over today's LSN.

### 3.1 Measurement Setup

**Video dataset:** As we aim to investigate the influences of network conditions on LVA performance, we use a fixed video dataset instead of live camera footage for repeatable experiments. Due to the lack of publicly available video datasets of high quality, high frame rate, and long duration, we use four YouTube videos (details shown in Table 2) as the source to create our video dataset. Specifically, for each video source, we extract a 480-second video clip and configure the client to read 1080p frames from the clip at a rate of 15 frames per second (FPS), thereby simulating a scenario where the frames are being captured in real time by a 1080p, 15 FPS camera.

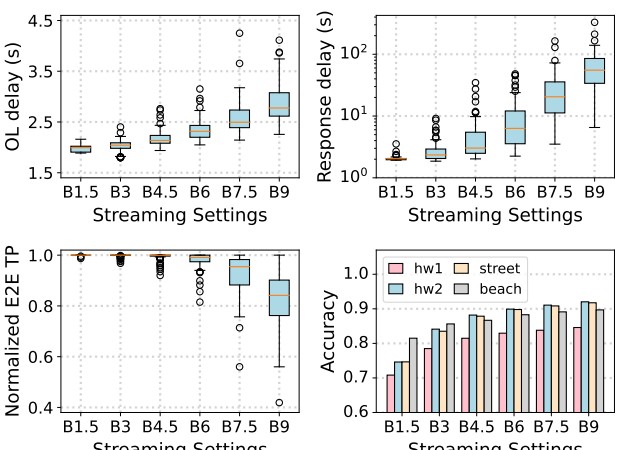

**Figure 2: In-the-wild LVA performance over Starlink.**

**Client and Server:** We choose the `aws-server`, a `p3.2xlarge` EC2 instance with an NVIDIA V100 GPU [3], as the analytics server and an off-the-shelf desktop as the Starlink UE.

**Server-side vision task and model:** We consider one basic video analytics task, object detection, as the server-side analytics task for proof-of-concept. In particular, the task is executed by a pre-trained DNN model, `YOLOv8l`, from the YOLOv8 family [27].

**Methodology:** By default, the H.264 codec is employed to compress raw video frames using a constant bitrate (CBR) encoding scheme, with a keyframe interval of 2 seconds. We explore target encoding bitrates of 1.5, 3, 4.5, 6, 7.5, and 9 Mbps, denoted respectively as `B1.5`, `B3`, etc. In addition to the encoding bitrate, LVA accuracy can also be affected by other encoding parameters, such as resolution and frame rate. Therefore, we evaluate a series of candidate frame rates {1, 3, 5, 15} and resolutions {1920 × 1080, 1280 × 720, 640 × 320} for each target bitrate. We then report the measurement results of the (frame rate, resolution) combination that yields the highest accuracy. For implementation, the client uses the `FFmpeg` C++ libraries [7] to encode and stream frames at the target bitrate, frame rate, and resolution. To further minimize the encoding and streaming latency, we set the encoder to the *ultrafast* and *zerolatency* mode [8], and only I-frames and P-frames are encoded in this mode. The server also uses the `FFmpeg` libraries to listen for stream events, receive streams, and decode the compressed video packets into uncompressed frames.

**Metrics:** We consider the following performance metrics.

◇ *Offloading delay (OL delay).* This measures the time it takes for frames to be encoded by the client, transmitted over the network, and decoded by the server. Specifically, a frame's offloading delay is the elapsed time from when the client starts encoding the frame to when it is prepared for analysis on the server. A GOP's offloading delay is the duration from when the client starts encoding its first frame to when the server completes the decoding of its last frame. The average offloading delay of all GOPs within a video is considered as the video's offloading delay.

◇ *Response delay.* A GOP's response delay is the total time elapsed from when the first frame of a GOP is captured to when the analysis results of the last frame in the GOP become available. It

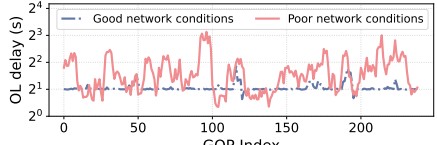

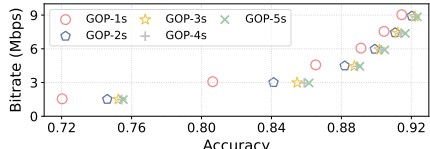

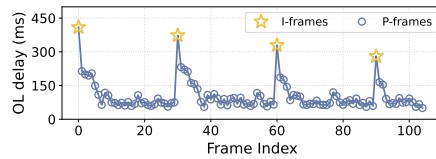

(a) Impact of network conditions (hw2, B6).

(b) Impact of GOP length in seconds (hw2).

(c) Impact of frame type (hw2).

Figure 3: In-depth examination of performance influencing factors.

includes delays caused by potential queuing, network delivery, and server-side model inference. The average response delay across all GOPs within a video is regarded as the overall response delay of the video.

◇ *Normalized end-to-end throughput (E2E TP).* Assume that the entire *capture-streaming-analysis* pipeline transmits and analyzes $n$ frames in $t$ seconds, i.e., the time elapsed from when the camera captures the first frame to when the server completes the analysis of the $n^{th}$ frame is $t$ seconds. Then, the normalized E2E TP is $n/(t \cdot f)$, where $f$ is the frame rate of the stream.

◇ *Accuracy.* Following the best practice in previous studies [14, 40], we regard the detection results of the same model on raw frames as ground truth to eliminate the inaccuracy caused by the object detection model itself. Accuracy is then calculated as the F1 score between the predicted and ground-truth results, as in [40]. A true positive is accepted when the intersection over the union between a predicted bounding box and a ground-truth bounding box is greater than 0.5, and their object categories are the same.

## 3.2 Measurement Results and Insights

We stream the videos under each setting at different times of the day for more than **10 days**. For all settings, the mean frame encoding delay on the client is about 15.83 ms, and the mean frame decoding delay on the server is about 3.73 ms. The server-side model inference delay for the highest resolution (i.e., $1920 \times 1080$) is about 62.01 ms on average. This means that the frame encoding, decoding, and model inference can all run at a speed exceeding 15 FPS, i.e., the maximum streaming frame rate. Thus, if any delays affect real-time analysis, it is reasonable to attribute them primarily to network transmission.

Figure 2 presents the statistical results of 20 experimental runs (or trials) conducted in the wild for each video-setting pair. As the offloading delay subfigure suggests, the higher the streaming bitrate is, the more delay is introduced during network transmission. With the streaming bitrate increasing beyond the capacity of the LSN, the offloading delay becomes the latency bottleneck in the entire processing pipeline, rendering it struggle to keep up with the frame arrival rate and leading to progressively accumulating lags. These lags eventually result in exponentially increasing response delays. Overall, it is still challenging for the LSN to support real-time LVA streaming (i.e., normalized E2E TP is 1.0) at bitrates higher than 6 Mbps. While satisfying the real-time requirements of low to medium bitrates seems promising, the resulting analytics accuracy can be significantly compromised. As a result, simultaneously achieving the low latency and high accuracy goals of LVA remains an open problem for today's LSN.

Another observation from Figure 2 is that the delay-related metrics vary significantly across multiple trials under identical streaming settings. For example, the response delay of video hw2 under setting B6 can be 2.49 seconds in one trial and 48.10 seconds in another, indicating a performance difference of 19.32×. Since the only variable between the trials is the underlying network conditions, this observation reveals the significant influence of the LSN performance on LVA-perceived QoE. *This calls for LSN-adaptive streaming solutions for LVA applications to provide consistent QoE.* Figure 3a further details the GOP offloading delay variations observed in these two trials. As shown, the offloading delay remains stable with only occasional small fluctuations under the good network conditions, suggesting that a coarse-grained adaptation strategy may be adequate. In contrast, under the poor network conditions, the offloading delay fluctuates wildly and frequently, necessitating a more nuanced, fine-grained adaptation approach. These findings uncover the challenges inherent in application-level LSN adaptation and motivate the design of granularity-variant adaptation strategies.

Given the self-contained structure, GOP is the natural encoding unit for bitrate control and network adaptation. We thus fix all the other variables and vary only the GOP length to investigate its impacts on the accuracy. Figure 3b shows that with the same target encoding bitrate, increasing GOP length can improve the analytics accuracy, and the improvements are particularly obvious at low target bitrates. We identify the reason is that at a given encoding bitrate, shorter GOPs result in a higher frequency of I-frames. This reduces the average frame size and increases the mean quantization parameter, leading to diminished overall image quality that adversely affects accuracy. Additionally, since a P-frame has to wait for its reference frame to arrive before it can be decoded at the server, the large size of an I-frame affects not only its own offloading delay but also the offloading delay of its subsequent P-frames as shown in Figure 3c. This observation encourages the adoption of longer GOP lengths, when network conditions permit, to benefit both analytics accuracy and delay stability.

## 4 TOWARDS LSN-ADAPTIVE LVA STREAMING

Inspired by our measurement insights, we propose StarStream, a novel adaptive LVA streaming framework specifically designed for LSNs. Figure 4 presents an overview of StarStream where the client continually streams captured frames over LSN to an analytics server. The client adapts video encoding and streaming GOP lengths and other configurations, such as bitrate, to the dynamic uplink network conditions for both accuracy and latency optimization. The core adaptation component is the *shift-guided*

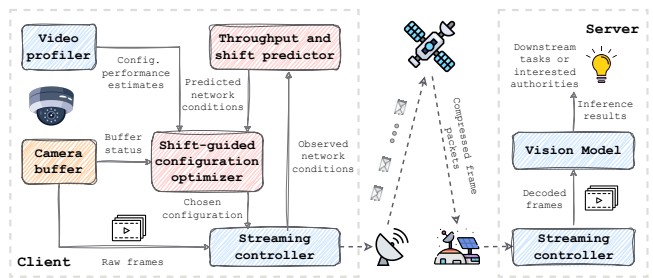

**Figure 4: Framework overview of StarStream.**

*configuration optimizer*, which decides how to encode and stream incoming video content according to predicted network throughputs and their shifts, up-to-date configuration performance, and camera buffer status, to strike a balance between accuracy and latency.

Specifically, the future network conditions are derived from the *throughput and shift predictor*, which predicts not only the future upload throughputs but also the timing of significant changes in throughput. The up-to-date configuration performance is estimated based on the information provided by the *video profiler*, which estimates the reference configuration performance by offline profiling representative frames and captures recent content characteristics by online compact model inference. The camera buffer status is obtained by querying the *camera buffer*, which caches captured video frames that cannot be promptly transmitted due to poor network conditions. After the *shift-guided configuration optimizer* makes the encoding configuration decision, the *streaming controller* component will encode and stream corresponding frames according to the chosen GOP length and configuration.

## 4.1 LSN Uplink Performance Prediction

Network throughput prediction serves as a foundational step in building network-adaptive multimedia applications. A multitude of throughput predictors based on historical observations (e.g., harmonic mean [34] and moving average [15]), classical machine learning models (e.g., decision tree [16] and random forest [1]), or deep learning models (e.g., fully connected network [32], long short-term memory [12], and Seq2seq [19]), have been developed for various terrestrial networks.

Compared to terrestrial networks, throughput prediction for LSN is much more challenging due to its inherent characteristics, e.g., vulnerabilities to exogenous factors. Recent years have witnessed the success of time series prediction models based on transformer [28], which have outperformed other time series prediction methods in various domains [30]. Their strengths in capturing long-term dependencies, modeling complex temporal interactions, and easily incorporating external information align well with the properties of an ideal uplink performance predictor for LSN. As a result, we propose a transformer-based throughput and shift predictor to fully utilize the uplink resources of LSN while mitigating its challenges. In particular, the predictor's design is based on Informer [44], an efficient transformer-based time series forecasting model, but with unique designs tailored to the characteristics of LSN.

Given the notable instability in LSN uplink performance, understanding the timing of significant throughput changes can help

make judicious decisions. For instance, if the throughput is predicted to remain stable in a future time horizon, we can choose a long GOP length to enhance accuracy. Conversely, if the throughput is expected to experience frequent variations, we can choose a short GOP length to allow for flexible configuration adjustments at GOP boundaries, thereby accommodating fine-grained throughput variations. As such, apart from a throughput prediction head, the proposed predictor also integrates a *throughput shift prediction* head, as shown in Figure 5. Formally, let $b_t$ denote the throughput at time step $t$. We define that a throughput shift occurs at time step $t$ if the difference between $b_t$ and $b_{t-1}$ is greater than a predefined threshold $\delta$. The *throughput shift prediction* head then outputs a binary shift indicator for each future time step, indicating whether a throughput shift is expected to occur. Both prediction heads are attached directly to the output layer of the Informer decoder.

As Figure 5 shows, the proposed predictor takes a sequence of network observations and their corresponding timestamps from time step $t - m + 1$ to $t$ as input, to predict the future network performance from time step $t + 1$ to $t + n$. Specifically, the Informer encoder receives the entire input sequence, while the decoder's input is crafted by concatenating two sequences: One is a subsequence of the input sequence, starting from time step $t - p + 1$ to time step $t$ (where $p \leq m$), and the other is the target sequence to be predicted, where unknown values are padded with zeros. Note that unlike traditional encoder-decoder style models that recursively generate the output for each time step one at a time, the decoder generates outputs for all time steps at once in a generative way.

Apart from the positional embedding that encodes the relative position information within the input sequence, the input to the predictor also integrates the outputs of the following three embedding layers. *1) Observable variables (OV) embedding layer:* This layer embeds observable variables for network performance prediction. In addition to the historical observations of throughput and its shifts, we also consider variables related to the underlying TCP connection, such as the retransmit times, the sending congestion window size, the smoothed RTT estimate, and the RTT variation, given their proven effectiveness on throughput prediction [16, 32]. *2) Date embedding layer:* Our measurements confirm that the wall-clock time can have certain influences on the LSN network performance (recall the peak and off-peak hours performance in §2). Consequently, the date embedding layer is introduced to encode the global time information. *3) Handover embedding layer:* Starlink schedules satellite-UE associations every 15 seconds, suggesting that handovers may occur as frequently as every 15 seconds [5, 24]. To account for the potential influence of handovers on network performance, a handover embedding layer is used to encode the current second's position in the 15-second scheduling window.

## 4.2 Shift-Guided Configuration Optimization

***GOP Length Selection:*** In our problem, the GOP length determines the encoding and decision-making granularity. Instead of setting the GOP length to a fixed value, StarStream's configuration optimizer dynamically adjusts the GOP length under the guidance of the predicted throughput shift indicators. For instance, assume that the interval between two consecutive time steps is one second, and the lookahead window size $n$ is 3. If the predicted throughputs and

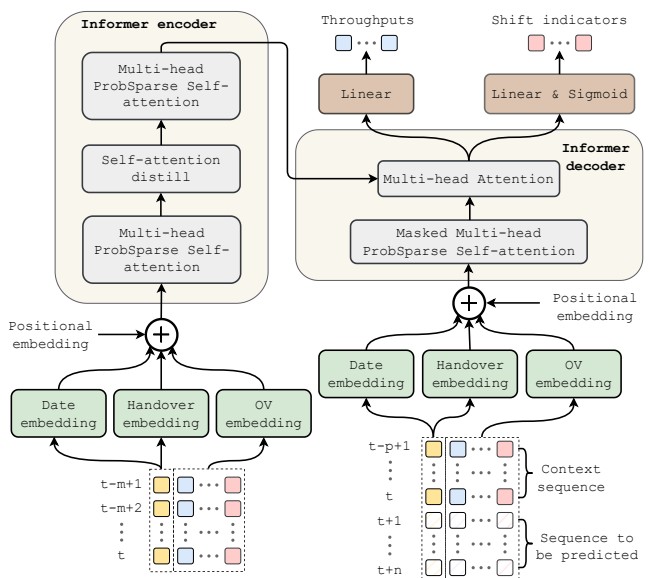

**Figure 5: Architecture of the proposed predictor.**

shift indicators are $[\hat{b}_{t+1}, \hat{b}_{t+2}, \hat{b}_{t+3}]$ and $[0, 0, 1]$, respectively, the chosen GOP length for the next GOP will be 2 seconds and the corresponding predicted throughput for the GOP will be calculated as $(\hat{b}_{t+1} + \hat{b}_{t+2})/2$. Then, with the chosen GOP length and the mean predicted throughput, the optimizer further chooses the encoding configuration (e.g., bitrate) for the GOP.

**Problem Formulation:** The performance goals of configuration selection are to maximize the analytics accuracy and minimize the lag, where lag is defined by the amount of queued-up frames that wait to be processed [41]. Without loss of generality, we assume that for the entire "capture-encoding-transmission-decoding-inference" processing pipeline, the primary cause of lag is network transmission. In continuous streaming scenarios, if the system's E2E processing speed can keep up with the frame capture speed, there will be no lags; otherwise, the newly captured frames will queue up at the camera buffer, resulting in an accumulation of lag. For example, if the frame capture speed is 10 FPS and the E2E processing speed is always 10 FPS, there will be no lags. However, if the E2E processing speed is always 5 FPS, the lag for the 1st, 2nd, and 3rd seconds will be 0.5 seconds, 1 second, and 1.5 seconds, respectively.

We quantify the lag using the camera buffer queue length and formulate the performance optimization problem from GOP $k1$ to GOP $k2$ ($k1 \leq k2$) as follows:

$$\underset{c_{k1}, \cdots, c_{k2}}{\arg\max} \quad \sum_{k=k1}^{k2} \alpha A_k(c_k) + \beta Q_k$$

$$s.t. \quad \begin{cases} \bar{b}_k = \frac{1}{t_k - t_{k-1}} \int_{t_{k-1}}^{t_k} b_t \, dt \\ t_k = t_{k-1} + \sum_j e_j(c_k) + \frac{\sum_j d_j(c_k)}{\bar{b}_k} + \Delta t_k \\ Q_k = Q_{k-1} + (t_k - t_{k-1}) - L_k \\ c_k \in C, \qquad \forall k = k1, \cdots, k2 \end{cases} \quad (1)$$

where $A_k(c_k)$ is the server-side analytics accuracy for GOP $k$ encoded with configuration $c_k$. $Q_k$ is the camera buffer queue length (in seconds) when the client finishes transmitting the last frame in GOP $k$. $\alpha$ and $\beta$ are positive weighting parameters used to trade off accuracy against lag. $t_k$ denotes the time when the client finishes transmitting the last frame of GOP $k$. $e_j(c_k)$ and $d_j(c_k)$ are the encoding delay and the compressed frame size of the $j^{th}$ frame in GOP $k$ encoded with configuration $c_k$, respectively. Note that in live streaming scenarios, frames are compressed and transmitted sequentially as they are captured. This means that the frame compression cannot be completed in advance; instead, compression and transmission occur in an interleaved manner. $\bar{b}_k$ is the average upload throughput between $t_{k-1}$ and $t_k$. $\Delta t_k$ denotes the total wait time between $t_{k-1}$ and $t_k$. A wait can happen when the frame upload speed is faster than the frame capture speed, and the client has to wait for the next frame to arrive to resume processing. $L_k$ is the chosen GOP length, and $C$ is the candidate configuration set.

**Content-Aware Configuration Performance Estimation:** Since we follow the common practice in live streaming video uploading and use CBR to encode videos [36], the data sizes and encoding delays of GOPs with the same length and configuration will not have significant differences. Thus, the *video profiler* component estimates $e_j(c_k)$ and $d_j(c_k)$ by offline profiling representative video content. The key challenge here is estimating $A_k(c_k)$, which is known to be related to video content dynamics. Existing solutions tend to combine offline and online profiling to adapt to changing video content [40]. However, online profiling requires sending raw frames to the server to acquire the ground truth analytics results for accuracy calculation. This can place a heavy burden on network transmission, especially when the video content is highly dynamic and online profiling needs to be conducted frequently. For networks like LSNs with very scarce uplink resources, the prohibitively high costs of online profiling can prevent it from being used in practice.

Let $A(c)$ denote the accuracy of configuration $c$ on the offline profiled video content. As the video content varies, using $A(c)$ to estimate the actual configuration accuracy may result in either an overestimate or an underestimate, depending on the relative analysis difficulty of the current content to the offline profiled content. Instead of directly updating the configuration's accuracy estimate, our workaround is to scale $A(c)$ by a factor of $\gamma$, which represents the relative analysis difficulty of the content being analyzed to the profiled video content.

To be specific, the *video profiler* runs a compact model (YOLOv8n by default) online to analyze newly captured frames for periodic estimate updates. The analysis difficulty of new content is inferred from the *confidence scores* output by the compact model. Intuitively, if the content is harder to analyze, there will be more detections with low confidence scores, indicating that the model is uncertain about these detections and more information is needed to ensure high accuracy. In particular, a detection is considered uncertain if its confidence score is lower than 0.5. We then define an uncertainty metric $u$, calculated as the ratio of the number of uncertain detections to the total number of detections. Let $u_n$ denote the uncertainty of new content and $u_p$ denote the uncertainty of the profiled content. With new content analyzed by the compact model, $\gamma$ is accordingly updated as $u_n/u_p$.

***Configuration Optimization to Balance Accuracy and Lag:***
Since the camera captures frames at a constant frame rate, the future frame capture time can be known in advance. Consequently, once the future upload throughputs and video-related variables are determined, $\Delta t_k$ can also be determined. This makes us ready for solving Problem (1). However, as relatively accurate prediction can only be guaranteed for a short future time horizon, the *optimizer* solves the problem over a finite time horizon (3 GOPs by default), following the model predictive control (MPC) [32, 34] paradigm.

In practice, evaluating every possible combination of frame rate, resolution, and bitrate can significantly prolong the online configuration optimization time. Fortunately, we analyzed the accuracy variations of all configurations and found that for all given bitrates, the best-performing (frame rate, resolution) combination is always one of three candidates, and their overall accuracies are very similar for a given bitrate. Considering the additional communication cost of changing the frame rate and resolution in the middle of the stream, we propose a profiling-based configuration pruning method. With the offline profiling results provided by the *video profiler*, it simply selects the (frame rate, resolution) combination that most frequently hits the top-3 performance under all candidate bitrates as the frame rate and resolution for online streaming. As such, the optimizer only needs to choose the bitrate for each GOP online. We further design an efficient dynamic programming (DP) algorithm to solve the optimization problem.

## 5 EVALUATION

### 5.1 Evaluation of Network Predictor

**Network Traces:** We use `iPerf3` to collect real-world LSN upload traces with the same client and server setup introduced in §3.1. The collected dataset comprises 504 network traces measured at different times of 17 days. Each trace has a duration of 10 minutes with a granularity of 1 second, including timestamps, upload throughput, and underlying TCP connection information. We also add a *shift* column to indicate whether a shift occurs, and the shift threshold $\delta$ is set to 2.5 Mbps. The dataset is further randomly divided into a training (70%) for predictor training, a validation set (10%) for model selection, and a testing set (20%) for model evaluation.

**Baselines:** We compare the proposed predictor with the following popular throughput prediction methods: harmonic mean (HM), moving average (MA), random forest (RF), fully connected network (FCN), long short-term memory (LSTM), and sequence-to-sequence (Seq2seq). Since these methods are designed to only predict the throughput, we calculate the throughput shift indicators by differencing the predicted throughputs and then comparing the differences to the shift threshold.

**Evaluation metrics:** Throughput prediction is a regression problem, so we consider the following metrics: Mean Absolute Error (MAE), Root Mean Squared Error (RMSE), and Mean Absolute Percentage Error (MAPE). Smaller values are better for these metrics. Since throughput shift prediction is essentially a binary classification problem, we use Accuracy and F1 score as the evaluation metrics. For both metrics, larger values are better.

**Result Analysis:** Table 3 presents the performance of different predictors, where the context sequence length $p$ of our method is 15. As shown, there exists a large performance gap between the

**Table 3: Comparison of different network predictors (lookback window size $m = 60$; lookahead window size $n = 15$) .**

| Methods | Throughput ↓ | | | Shift indicator ↑ | |
|---|---|---|---|---|---|
| | MAE | RMSE | MAPE | Accuracy | F1 |
| HM [34] | 4.019 | 5.275 | 57.095 | 0.670 | 0.074 |
| MA [15] | 3.166 | 4.045 | 52.173 | 0.671 | 0.065 |
| RF [1] | 2.577 | 3.388 | 42.695 | 0.682 | 0.025 |
| FCN [32] | 2.493 | 3.302 | 41.042 | 0.684 | 0.040 |
| LSTM [12] | 2.472 | 3.281 | 40.513 | 0.684 | 0.041 |
| Seq2seq [19] | 2.463 | 3.274 | 40.383 | 0.685 | 0.053 |
| Ours | 2.435 | 3.248 | 39.244 | 0.706 | 0.467 |

two naive prediction methods (i.e., HM and MA) and other methods. This implies that simple models relying on historical throughput information can hardly capture the complex variations in LSN upload throughput. RF, FCN, and LSTM are able to capture the complex relationships between input features and hit competent performance. However, they treat the throughput at different future time steps equally as output features to be predicted and may not fully exploit the temporal relationships between multi-step outputs. Seq2seq overcomes this deficiency by using a decoder to recursively predict the output at each time step and achieves better performance.

Our predictor achieves the best performance on all evaluation metrics. This can be attributed to the introduction of the attention mechanism and embeddings of exogenous information. Additionally, the performance of using predicted throughputs to estimate the throughput shifts is notably worse. We find the reason is that prediction models tend to generate *smoothed* multi-step throughput forecasts, which renders the shift indicator rarely equal to 1. This also verifies our design advantages of allowing the model to directly predict the shift indicators.

### 5.2 Evaluation of StarStream

**Methodology:** We use the video dataset shown in Table 2 for performance evaluation. We consider the same frame rate, resolution, and bitrate candidates as in §3.1, and 5 GOP length candidates, i.e., {1, 2, 3, 4, 5} seconds. In the offline stage, the *video profiler* profiles the first 20-second of each video to obtain each configuration's performance and processing costs (including encoding/decoding/inference delays and compressed frame sizes). Then, it selects the streaming frame rate and resolution for each video based on the proposed pruning method. The update of configuration accuracy estimates is conducted every 30 seconds online to profile 5 seconds of newly captured video content. For a fair comparison of different solutions, we implement a trace-driven simulator based on the test network traces introduced in §5.1 and video processing traces collected offline. Specifically, we encode and stream each video with different configurations and GOP lengths from the client to the server (same as that in §3.1), while recording the corresponding compressed frame sizes, encoding delays, and decoding/inference delays, to construct the video processing traces dataset. The traces for the same video are well aligned to facilitate flexible GOP and configuration switching during the online simulation stage. $\alpha$ and $\beta$ in Problem (1) are set to 1 and 0.02 by default, respectively.

**Figure 6: Overall performance of different solutions. CDF figures are statistics on all video-trace pairs.**

**Baselines:** All of the baselines use a fixed GOP length of 2 seconds.

◇ `Fixed`: This non-adaptive solution streams videos at a fixed bitrate, selected as the highest bitrate that is below the mean throughput observed 1 minute before the streaming begins.

◇ `AdaRate`: This is a pure rate-based adaptive streaming solution. It employs the same network predictor as `StarStream` but simply chooses the maximum bitrate below the predicted throughput.

◇ `MPC`: This solution uses MPC [32, 34] to optimize the objective of Problem (1) over 3 future GOPs. The future throughputs are estimated using harmonic means of past 5 GOPs. The video-related variables are estimated by offline profiling the first 20-second video content.

**Performance Metrics:** We consider the *accuracy*, *normalized E2E TP*, *OL delay* and *response delay* defined in §3.1 as the metrics. Note that since different videos can be streamed at different frame rates and the GOP length can also change within a stream, the OL delay and response delay in this evaluation are uniformly defined for per-second video content, rather than for per frame or GOP.

**Performance Improvement:** Figure 6 shows the overall performance of different methods. As shown, `Fixed` cannot achieve real-time E2E processing (i.e., normalized E2E TP is 1.0) for most video-trace pairs because this rigid method cannot adapt to the ever-changing network conditions. In comparison, `AdaRate` improves the overall normalized E2E TP, OL delay, and response delay by dynamically adjusting each GOP's bitrate based on the predicted network throughput. However, it inevitably suffers from performance degradation due to imperfect predictions. In addition, it has no mechanism to automatically recover from the previously made bad decisions, which can lead to accumulated lags that eventually reduce the normalized E2E TP and increase the response delay.

By integrating the network predictions along with the camera buffer queue status, both `MPC` and `StarStream` achieve near real-time E2E processing for almost all video-trace pairs. By strategically optimizing over multiple future GOPs to trade-off between accuracy and lag, these methods can timely recover from previously made bad decisions and control the response delays in a reasonable range (i.e., < 10 seconds). `StarStream` further benefits from the more flexible GOP length configuration and more accurate configuration accuracy estimation, achieving noticeable accuracy improvements over the `MPC` solution.

**Ablation Study:** We consider two variants: One disables the online configuration accuracy updates, named V1, and the other replaces the network predictor with a Seq2seq predictor, named V2. We find that, compared to `StarStream`, the mean response delay of V1 increases by 8.6% while the accuracy remains almost the same, and the mean response delay of V2 increases by 10.1% while the

mean accuracy decreases from 0.867 to 0.864. This confirms the effectiveness of the key components of `StarStream`.

**System Overheads:** The system overhead of `StarStream` stems from three primary sources: the predictor, the DP algorithm, and the online content profiling. We benchmark these overheads using our measurement client, equipped with a 6-core Intel i7-6850K CPU and an Nvidia GeForce GTX 1080 Ti GPU. Notably, the DP algorithm is highly efficient, solving the problem in just $0.63 \pm 0.35$ ms on the CPU. It takes about 1.44 seconds to profile 5-second raw frames on the GPU, and the fast profiling speed can ensure that the profiled video content is fresh. The inference delay of the predictor is about $13.0 \pm 5.1$ ms on the GPU, and $20.0 \pm 1.4$ ms on the CPU. In practice, we can schedule online video profiling and network prediction to run in parallel with video streaming so that they do not introduce additional delays to the streaming pipeline.

# 6 RELATED WORK

Efficiently streaming videos over networks for online analytics has been a hot research topic in recent years [13–15, 20, 40, 42]. Shipping massive video data over networks in real time requires tremendous bandwidth resources, which places a heavy burden on existing network infrastructures. Hence, various techniques have been proposed to reduce the amount of offloaded data without significantly compromising accuracy, such as controlling video encoding knobs (e.g., resolution) [40], frame filtering [14], frame masking [15], and frame partitioning [42]. These are all valuable attempts to build high-performance LVA systems but are discussed in the context of terrestrial networks. Terrestrial network conditions between cameras and servers are relatively stable. Even with the scarce and dynamic wide area network links, simple network probing and bandwidth estimation methods can be effective for adaptation [40]. To the best of our knowledge, we are the first to discuss building high-performance LVA applications over LSN.

# 7 CONCLUSION

LVA enables machines to glean information from the physical world in real time. It is poised to become the driving force behind advanced intelligence. This paper closely investigated the performance of the emerging LSN in supporting LVA applications. Through extensive measurements, we found that the uplink resources of the LSN are still scarce, and the wild fluctuations in network conditions can prevent upper-level applications from providing consistently satisfactory QoE. We further proposed a novel adaptive LVA streaming framework, `StarStream`, to improve the performance of LSN-enabled LVA applications. Extensive trace-driven experiments verified the effectiveness and superiority of our proposed solution.

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
