# OpenReview forum: "StarStream: Live Video Analytics over Space Networking"
_acmmm.org/ACMMM/2024/Conference — MM2024 Oral_

### Official Review · Reviewer_N2gb · 2024-05-23

**Rating:** 4
**Confidence:** 2

**Summary:**

The paper "StarStream: Live Video Analytics over Space Networking" explores the use of Low Earth Orbit (LEO) satellite networks, specifically Starlink, to enhance live video analytics (LVA) capabilities in remote and disaster-stricken areas where terrestrial networks fall short. The authors identify significant challenges, such as uplink bottlenecks and volatile network conditions in LEO networks. To address these issues, they propose StarStream, a novel streaming framework that adapts to network conditions using a transformer-based network performance predictor and a content-aware configuration optimizer. Extensive measurements and trace-driven experiments demonstrate that StarStream significantly improves the performance and reliability of LVA over LEO satellite networks.

**Strengths:**

The paper's main strengths are its novel application of LEO satellite networks to live video analytics, its robust theoretical approach, comprehensive evaluation, and clarity of presentation.
- The paper employs a transformer-based model for network performance prediction, leveraging its ability to handle sequential data and capture dependencies over time. Given LEO satellite networks' complex and dynamic nature, this choice is theoretically sound. The network performance predictor is described with detailed architecture, incorporating elements like positional embedding, observable variables embedding, date embedding, and handover embedding, which are all relevant to the prediction task.
- The authors present a detailed evaluation setup, including measuring network performance metrics and using a diverse video dataset from YouTube to simulate real-world scenarios. The evaluation metrics are well-chosen, focusing on offloading delay, response delay, normalized end-to-end throughput, and accuracy. These metrics comprehensively cover the performance aspects critical to live video analytics.
- The paper discusses the practical implications of the StarStream framework, highlighting its potential to transform live video analytics in challenging environments. The applications mentioned, including disaster response and wildlife monitoring, underscore the framework's relevance and potential impact.

**Limitations:**

- Dataset Limitations: The paper’s evaluation relies on four YouTube videos, which may not comprehensively represent the wide variety of conditions that affect the performance of the video profiler component.
- To comprehensively assess regression methods, incorporating additional metrics such as R2 values, scatter plots, and the standard deviation of absolute errors can offer a deeper understanding of model performance.
- Table 3 showcases that the outcomes yielded by the proposed approach marginally outperform those of alternative methods. However, one must weigh whether this incremental improvement justifies the investment in time and resources associated with employing attention-based models.
-  The selection process for the look-back window size appears to lack clarity within the study. It would be beneficial if the authors provided insight into whether the choice of window size was determined through experimentation during model training.
- In this analysis, the thorough examination of the LSN fluctuation stands out, yet the treatment of the results and ablation studies appears relatively brief. While the detailed exploration of LSN fluctuations is commendable for understanding system dynamics, the brevity of the discussion on results and ablation studies potentially limits the depth of insight into the effectiveness and robustness of the proposed approach.

**Suitability:**

3

---

### Official Review · Reviewer_AB7f · 2024-05-25

**Rating:** 4
**Confidence:** 2

**Summary:**

Authors analyse the possibility of leveraging low orbit (LEO) satellite networks for live video analytics (LVA). For the experiments, authors choose Starlink, performing real-world experiments. The main findings show significant negative impact of uplink bottleneck and volatile network conditions on LVA performance. To counter drawbacks posed by current satellite systems, authors propose a StarStream, a novel LVA transformer-based framework for adapting length of GOP striking a balance between classification accuracy and latency.

**Strengths:**

- Paper is well written, and the problem formulation is clear.
- Experiments in the wild are exhaustive and provide a solid foundation for the problem analysis
- Balancing between length of GOP pictures and induced latency is a interesting and challenging problem
- While results for throughput accuracy are similar between LSTM, Seq2Seq and proposed model, further experiments together with ablation study indicate importance of the proposed method and positive impact on overall QoE

**Limitations:**

- Authors don't discuss the main issue with deep learning models, namely, data distribution change and its impact on the effectiveness of the model. Please provide possible solutions to the challenge of distribution shift
- I am not convided by ablation study, considering very similar results related to throughput prediction accuracy between three deep learning models. Please, provide a more in depth analysis and explanation for the obtained results.

**Suitability:**

3

---

### Official Review · Reviewer_wKjD · 2024-05-27

**Rating:** 4
**Confidence:** 3

**Summary:**

The study "StarStream: Live Video Analytics over Space Networking" is motivated by the limitations of existing live video analytics (LVA) systems built over terrestrial networks, which restrict their applications in remote areas and during natural disasters. The key contributions include conducting extensive in-the-wild measurements using Starlink to gain insights into the achievable LVA performance over Low Earth Orbit (LEO) satellite networking (LSN) and developing StarStream, a novel LSN-adaptive streaming framework for LVA. StarStream is "empowered by a transformer-based network performance predictor tailored for LSN and a content-aware configuration optimizer." The authors demonstrate the effectiveness and superiority of StarStream through trace-driven experiments with real-world network and video processing data, highlighting its potential to enable high-quality LVA applications over LSN in challenging environments.

**Strengths:**

Authors make contributions to the field of multimedia systems research by addressing the unique challenges of live video analytics (LVA) over Low Earth Orbit (LEO) satellite networks (LSNs). The paper's claims are well-supported by extensive
empirical evaluations and real-world measurements using Starlink as a testbed, which reveal the impact of uplink bottlenecks and network volatility on LVA performance in LSNs. The key strength of this work lies in the development of StarStream, an adaptive LVA streaming framework specifically designed to tackle the distinct characteristics of LSNs. StarStream incorporates novel components, such as a
transformer-based network performance predictor and a content-aware configuration optimizer, which introduce solutions to enhance LVA performance over space networks. The effectiveness of StarStream is validated through trace-driven experiments, demonstrating improvements in accuracy and latency compared to existing approaches. This research is relevant to the ACM Multimedia community as it combines advancements in video analytics, network performance prediction, and adaptive streaming, pushing the boundaries of LVA deployment to challenging environments enabled by satellite networks.

**Limitations:**

There are several limitations that need to be addressed to strengthen the paper's contributions.

Firstly, The network traces used for training and evaluating the predictor are collected over 17 days. This may not capture the full range of variability in LSN performance across different times of the year, weather conditions, geographies etc. A more comprehensive dataset covering a longer time period would strengthen the evaluation. If these 17 days were spread over the entire year, it would be helpful to provide that information in the manuscript.

Secondly, For the content-aware configuration performance estimation, using the analysis confidence scores of a compact model as a proxy for the full model's accuracy on that content is an oversimplification. Low confidence doesn't necessarily mean low
accuracy. A more rigorous accuracy estimation approach should be considered.

Thirdly, the paper's evaluation relies on a limited video dataset from YouTube, which may not represent the diversity and complexity of real-world LVA scenarios. Either use a larger set of videos, or justify the selection of the videos used in the study.

From a novelty perspective, while StarStream tackles the important problem of LVA over LSNs, the machine learning contributions, such as the transformer-based predictor and content-aware optimizer, seem incremental compared to prior work. The paper would benefit from a more detailed literature review, identifying gaps in existing methodologies and explicitly stating how StarStream addresses them.

**Suitability:**

3

---

### Meta-Review · Area_Chair_Mbhx · 2024-07-04

**Recommendation:** Accept (Oral)
**Confidence:** 4

**Metareview:**

Pros

- Well-written and clear manuscript
- Novel idea and clear problem statement

Cons

- Short-duration traces used for training and evaluating the predictor
- Limited video dataset from YouTube
- Convincing rebuttal